# Unequal exchange of labour in the world economy

Jason Hickel [1,2,3] ✉, Morena Hanbury Lemos [1] ✉ & Felix Barbour [4,5] ✉

Researchers have argued that wealthy nations rely on a large net appropriation of labour and resources from the rest of the world through unequal exchange in international trade and global commodity chains. Here we assess this empirically by measuring flows of embodied labour in the world economy from 1995–2021, accounting for skill levels, sectors and wages. We find that, in 2021, the economies of the global North net-appropriated 826 billion hours of embodied labour from the global South, across all skill levels and sectors. The wage value of this net-appropriated labour was equivalent to €16.9 trillion in Northern prices, accounting for skill level. This appropriation roughly doubles the labour that is available for Northern consumption but drains the South of productive capacity that could be used instead for local human needs and development. Unequal exchange is understood to be driven in part by systematic wage inequalities. We find Southern wages are 87–95% lower than Northern wages for work of equal skill. While Southern workers contribute 90% of the labour that powers the world economy, they receive only 21% of global income.

Scholars of international political economy have argued that growth and capital accumulation in the wealthy 'core' states of the global North relies on the appropriation of value—labour, resources and goods—from the 'peripheries' and 'semi-peripheries' of the global South[1–8]. In the contemporary world economy, this appropriation occurs in large part through what scholars have defined as 'unequal exchange' in international trade[9–12]. Literature in this field has described how core states and firms leverage their geopolitical and commercial power to compress wages, prices and profits in the global South, both at the level of national economies as well as within global commodity chains (which account for more than 70% of trade), such that Southern prices are systematically lower relative to Northern prices[13,14]. Price inequalities compel Southern states and producers to export more labour and resources embodied in traded goods to the global North each year in order to pay for any given level of imports, enabling Northern economies to net-appropriate value to the benefit of Northern capital and consumers.

Dynamics of unequal exchange are understood to have intensified in the 1980s and 1990s with the imposition of structural adjustment programmes (SAPs) across the global South[15]. SAPs devalued Southern currencies, cut public employment and removed labour and environmental protections, imposing downward pressure on wages and prices. They also curtailed industrial policy and state-led investment in technological development and compelled Southern governments to prioritise 'export-oriented' production in highly competitive sectors and in subordinate positions within global commodity chains[1,16–18]. At the same time, lead firms in the core states have shifted industrial production to the global South to take direct advantage of cheaper wages and production costs, while leveraging their dominance within global commodity chains to squeeze the wages and profits of Southern producers[13,14,19]. These interventions have further increased the North's relative purchasing power over Southern labour and goods[15].

Several studies have sought to quantify the scale of appropriation through unequal exchange indirectly by adjusting monetary trade

[1]Institute for Environmental Science and Technology (ICTA-UAB), Autonomous University of Barcelona, Barcelona, Spain. [2]Department of Anthropology, Autonomous University of Barcelona, Barcelona, Spain. [3]International Inequalities Institute, London School of Economics and Political Science, London, United Kingdom. [4]Stockholm Resilience Centre, Stockholm University, Stockholm, Sweden. [5]Beijer Institute of Ecological Economics, The Royal Swedish Academy of Sciences, Stockholm, Sweden. ✉e-mail: j.e.hickel@lse.ac.uk; marianamorena.hanbury@uab.cat; felix.barbour@su.se

volumes for North–South disparities in wages[9,20] or general prices[15,21]. More recent research has used environmentally extended multi-regional input-output (EEMRIO) models, which enable us to track the flows of resources embodied in each nation's final consumption. These studies demonstrate empirically that the core economies rely on a physical net appropriation of embodied labour and resources from the global South[22–24]. However, this research has so far not directly analyzed price dynamics associated with the labour time embodied in North–South trade. It has also been unable to answer questions about the extent to which North–South wage disparities and unequal exchange may be an effect of differences in the type of labour being performed, such as in terms of skill level or sector (for instance, if wage inequalities arise because the South trades low-skilled labour for high-skilled labour, or primary goods for secondary goods).

In this study, we use the EEMRIO model EXIOBASE to track flows of embodied labour between North and South, for the first time accounting directly for sectors, wages and skill levels (as defined by the International Labour Organisation, ILO, described in Methods). This enables us to define the scale of labour appropriation through unequal exchange in terms of physical labour time, while also representing it in terms of wage value, in a manner that accounts for the skill level composition of labour embodied in North–South trade. Our category for the global North approximates the IMF list of 'advanced economies', with the South comprising all emerging and developing economies (see Methods). All monetary units are in constant 2005 Euros, corrected for inflation, represented in market exchange rates (MER), which is appropriate for international comparisons of income purchasing power in the global economy (see Methods).

We arrive at several major conclusions. (1) We find that the labour of production in the world economy, across all skill levels and all sectors, is overwhelmingly performed in the global South (on average 90–91%), but the yields of production are disproportionately captured in the global North. (2) The North net-appropriated 826 billion hours of embodied labour from the global South in 2021 (in other words, net of trade). This net appropriation occurs across all skill categories and sectors, including a large net appropriation of high-skilled labour. (3) The wage value of net-appropriated labour was €16.9 trillion in 2021, represented in Northern wages, accounting for skill level. In wage-value terms, the drain of labour from the South has more than doubled since 1995. 4) North–South wage gaps have increased dramatically over the period, across all skill categories and sectors, despite a small

improvement in the South's *relative* position. Southern wages are 87–95% lower than Northern wages for work of equal skill as of 2021, and 83–98% lower for work of equal skill within the same sector. (5) Workers' share of GDP has generally declined over the period, by 1.3 percentage points in the global North and 1.6 percentage points in the global South.

## Results

### Contributions to global production

We find that, in 2021, the final year of data, 9.6 trillion hours of labour went into producing for the global economy. Of that, 90% was contributed by the global South (Fig. 1). The South contributed the majority of labour across all skill levels: 76% of all high-skilled labour, 91% of medium-skilled labour and 96% of low-skilled labour. In the same year, 2.1 trillion hours of labour went into the production of internationally traded goods (our use of 'traded goods' in this paper refers to both goods and services). The relative North–South contribution to the production of traded goods is similar to that of total production, with the South contributing 91% of all labour (73% of all high-skilled labour, 93% of medium-skilled labour, and 96% of low-skilled labour). Note the latter figures are underestimates, given that most global South countries are aggregated into regions in EXIOBASE (see Supplementary Table 1) and trade within these regions is not represented.

The South's contribution to total global production has increased steadily over the period since 1995, across all skill categories. The largest increase has occurred in the high-skill category, with the South's contribution to high-skill production increasing from 66% of the world's total in 1995 (1.9x more than the North) to 76% in 2021 (3.2x more than the North). In fact, the South now contributes more high-skilled labour to the world economy (1124 billion hours in 2021) than all the high-, medium- and low-skilled labour contributions of the global North combined (971 billion hours in 2021). The South also contributes the overwhelming majority of labour across all aggregated sector groupings we derived from EXIOBASE, including agriculture (99%), mining (99%), manufacturing (93%), services (80%) and 'other' (89%). See Methods for sector aggregations.

Despite contributing 90–91% of the total labour that goes into global production and the production of traded goods in 2021, including the majority of high-skilled labour, the global South received less than half (44%) of global income, and Southern workers received

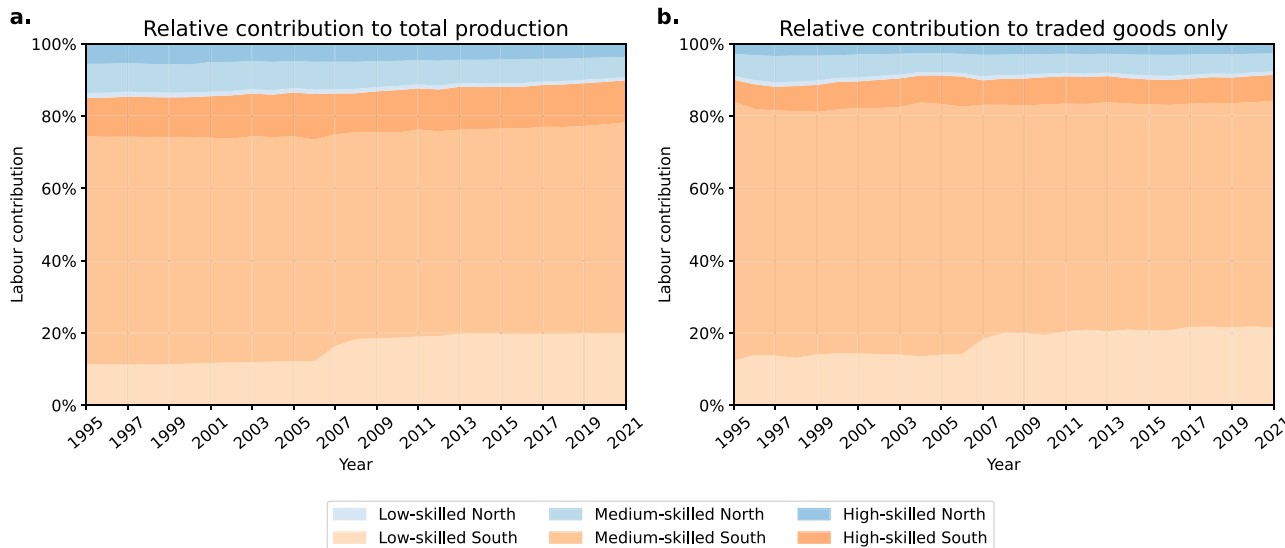

**Fig. 1 | Relative contributions of labour (hours) to global production by region and skill level, 1995–2021.** Blue indicates labour rendered by workers in the global North, while orange indicates labour rendered by workers in the global South. Skill levels are shaded from lighter (low-skilled) to darker (high-skilled). Panel **a** shows contributions to total production of all goods and services. Panel **b** shows contributions to production of traded goods and services only.

only 21% of global income in that year. In other words, while global production is overwhelmingly performed in the global South, the yields are disproportionately captured in the global North, indicating a disproportionate command of the global product.

Table 1 shows that the total number of employed workers and total number of hours worked has increased in both the North and the South from 1995 to 2021, with the increase substantially larger in the global South. The final rows illustrate several interesting points. First, we see that workers in the global South consistently render more labour per worker than in the North, by large margins. In the final year of data, Southern workers worked on average 466 h more than their Northern counterparts (26% more). Second, we see that in the North, labour time per worker has decreased by 7% over the period, while in

the South it has increased by 1%. To the extent that increased labour time has contributed to global economic growth over the past 25 years, this burden has been shouldered overwhelmingly by people in the global South.

## Unequal exchange of labour

Our analysis confirms a substantial and persistent pattern of unequal exchange between the global North and South. In 2021, the global North imported 906 billion hours of embodied labour from the South while exporting only 80 billion hours in return (a ratio of 11:1). On average across the period, the North imported 15x more labour from the South than it exported in return. In other words, the North net-appropriates large quantities of labour from the South. This net appropriation occurs across all skill categories, including high-skilled labour. On average the North imports 4x more high-skilled labour from the South than it exports, together with 17x more medium-skilled labour and 29x more low-skilled labour. Figure 2 shows labour exports and imports by the global South across the period 1995–2021.

The unequal exchange of labour described above is not explained by sectoral differences. We found that the global North net-imports large quantities of labour from the South in all skill levels across all five sectors. On average, the North imported 120x more agricultural labour than it exported, 110x more mining labour, 11x more manufacturing labour, and 6x more service labour. In other words, it is not the case that the North net-imports labour in primary production from the South while net-exporting a smaller quantity of labour in secondary and tertiary production. On the contrary, the global North relies on a net appropriation of labour from the South across all sectors,

**Table 1 | Workers, labour time, and labour intensity by region, in 1995 and 2021**

|  | 1995 | 2021 | Change |
|---|---|---|---|
| **Total number of workers employed (thousands)** |  |  |  |
| North | 413,690 | 548,729 | +33% |
| South | 2,007,495 | 3,881,799 | +93% |
| **Total labour rendered (millions of hours)** |  |  |  |
| North | 783,962 | 971,234 | +24% |
| South | 4,448,689 | 8,677,863 | +95% |
| **Labour time per worker (hours)** |  |  |  |
| North | 1895 | 1770 | −7% |
| South | 2216 | 2236 | +1% |

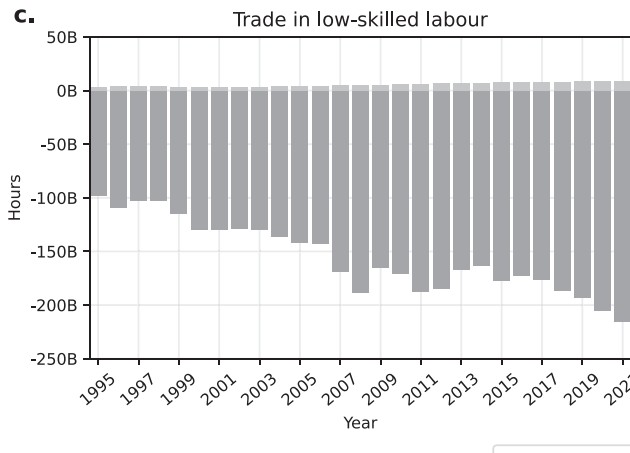

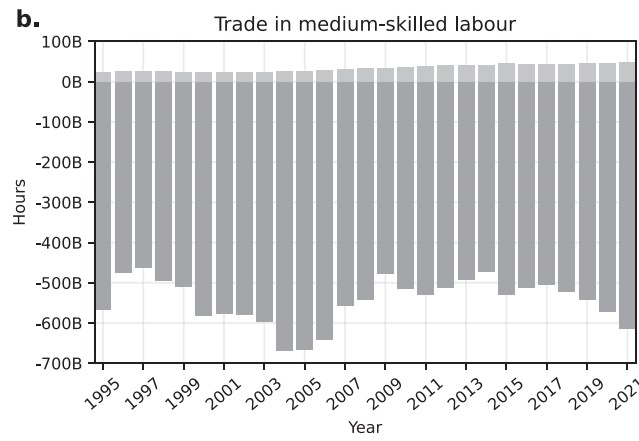

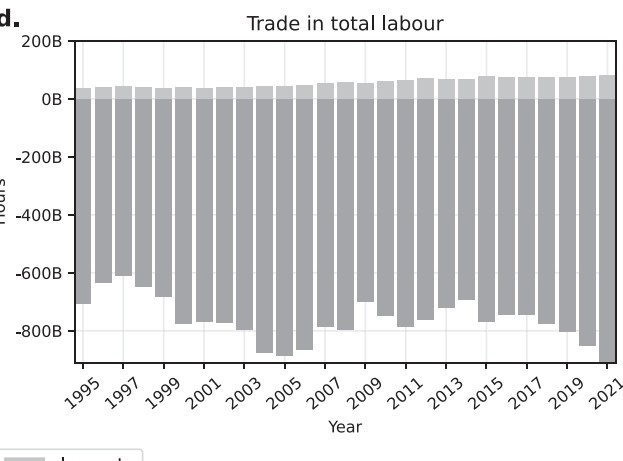

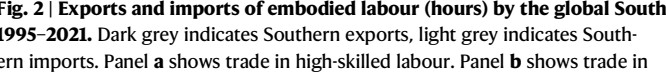

**Fig. 2 | Exports and imports of embodied labour (hours) by the global South, 1995–2021.** Dark grey indicates Southern exports, light grey indicates Southern imports. Panel **a** shows trade in high-skilled labour. Panel **b** shows trade in medium-skilled labour. Panel **c** shows trade in low-skilled labour. Panel **d** shows trade in total labour across all skill-levels (the sum of **a**–**c**).

including manufacturing and services. There is no sector in which the North net-exports labour to the South. This is demonstrated in Supplementary Fig. 1.

The time-series results demonstrate that the global South's position deteriorated during 1995-2005, with the South–North exchange ratio increasing from 17:1 in 1995–97 to 21:1 in 2003–2005. During this period, which was characterised by draconian structural adjustment policies applied during the 80s and 90s, Southern economies were compelled to increase their exports of embodied labour by 24% simply in order to maintain the same quantity of imports from the North. The South's position improved over the following decade (2005–2015), as the most aggressive adjustment policies were loosened and as the commodity boom took off, with the exchange ratio dropping to 10:1.

This improvement was driven predominantly by an improvement in the position of China. However, improvements have ceased since 2015, and some regression has occurred.

Figures 3 and 4 show the total quantity of labour net-appropriated by the North over the period, by skill level and sector, respectively. The North net-appropriates labour across all skill levels and all sectors. It is worth noting that appropriation in the secondary and tertiary sectors (manufacturing and services) is now greater than in the primary sectors (agriculture and mining), and indeed this has been the case for most of the period covered.

The total net appropriation increased from 1995 to a peak in 2005 before declining during the decade 2005–2015. We find that the decline during 2005–2015 corresponds with an improvement in

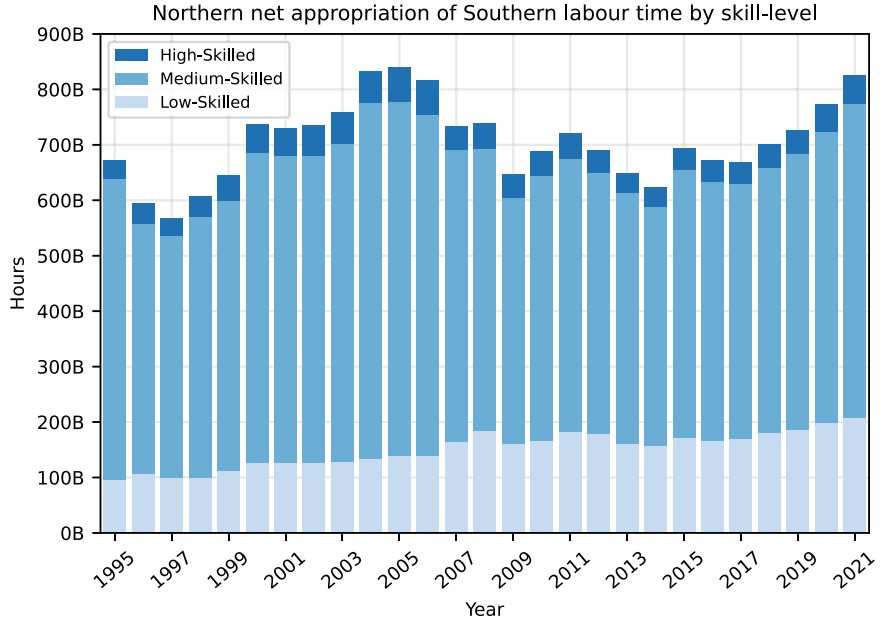

**Fig. 3 | Northern net appropriation of labour (hours) from the global South, by skill level, 1995–2021.** Skill levels are shaded from lighter (low-skilled) to darker (high-skilled). These figures correspond to the net of imports and exports represented in Fig. 2.

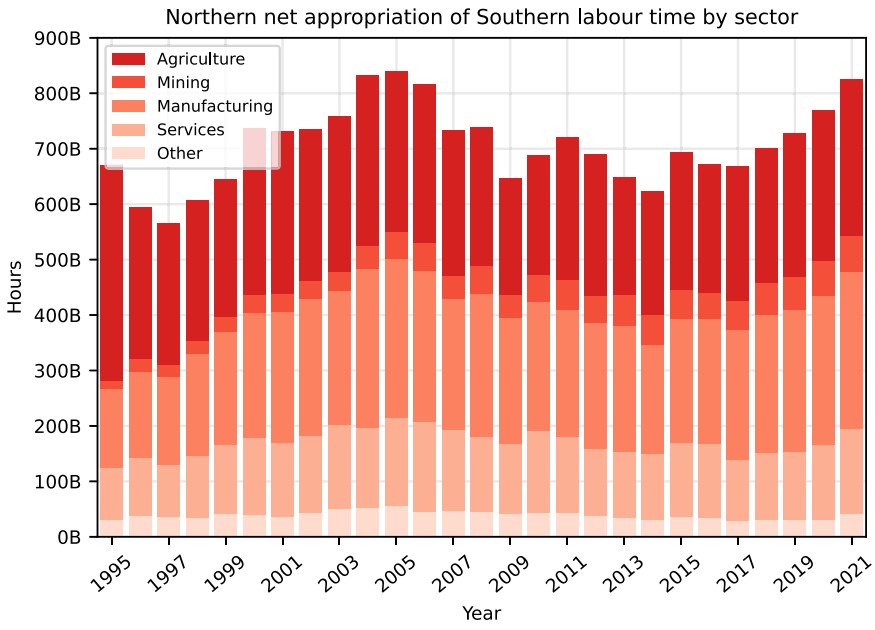

**Fig. 4 | Northern net appropriation of labour (hours) from the global South, by sector, 1995–2021.** The figure shows labour net-appropriated in agriculture, mining, manufacturing, services and 'other' (defined in Methods).

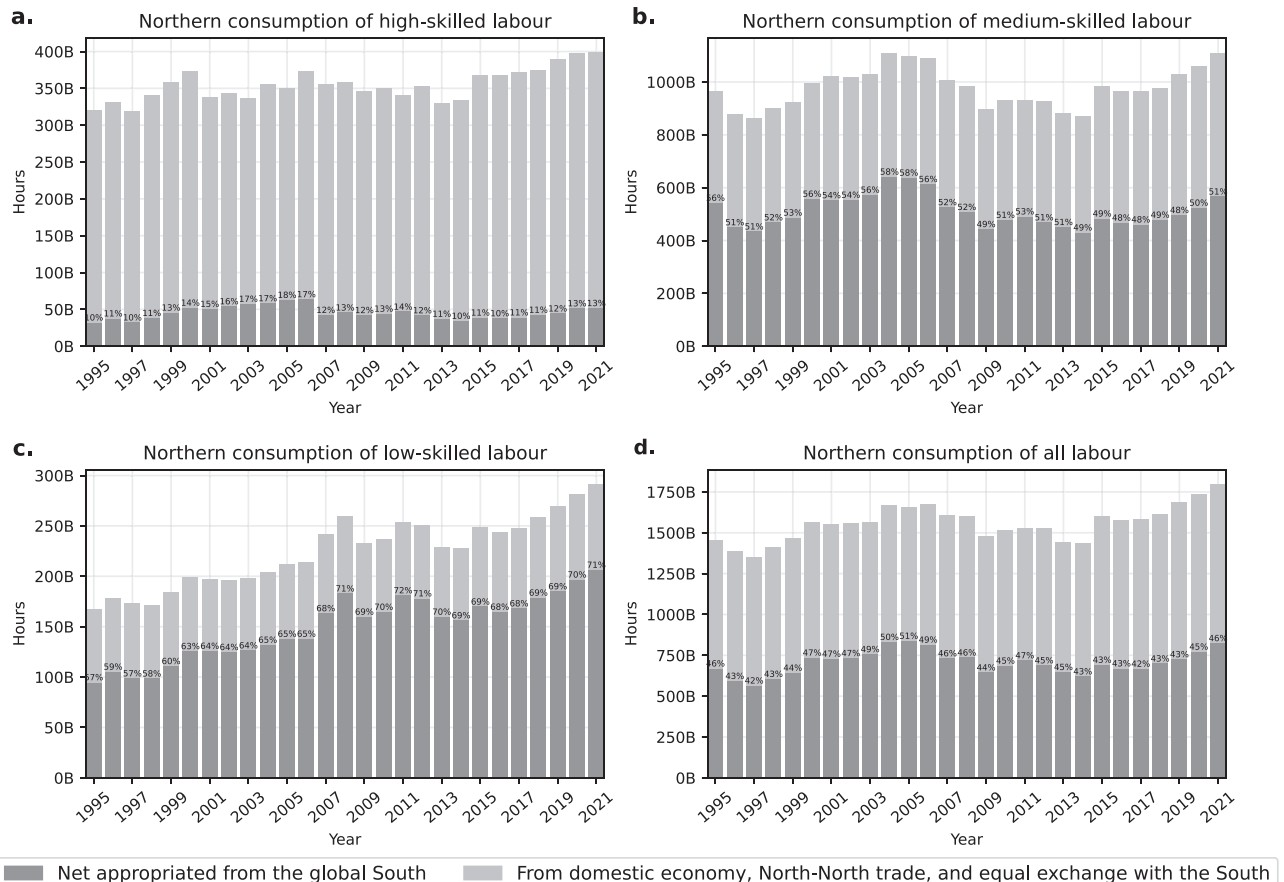

**Fig. 5 | Northern net appropriation of labour (hours) from the South as a share of Northern consumption of labour.** The dark grey bars represent labour net-appropriated from the global South (corresponding to the figures represented in Fig. 3). The light grey bars represent labour consumed in the North from all other sources: from the domestic workforce, from North-North trade, and from equal exchange with the South. Panel **a** shows Northern consumption of high-skilled labour. Panel **b** shows Northern consumption of medium-skilled labour. Panel **c** shows Northern consumption of low-skilled labour. Panel **d** shows Northern consumption of all labour (the sum of **a**–**c**).

Southern wages against Northern wages. However, this improvement ceased in 2015 and wage ratios have stabilised (see section titled "Wage trends" below). The appropriation has increased in the years since, driven by an increase in the volume of trade. In 2021 the total quantity of net appropriation reached 826 billion hours. These figures are substantially larger than previous studies have found (see Supplementary Discussion 3 for details)[25]. Net flows from China to the North account for roughly one-sixth of total net South–North flows. It is worth noting here that the net South–North flow of embodied labour is not 'paid for' by an opposite net flow of embodied land, energy or materials (on the contrary, large net South–North flows occur across all input categories).

We find that this pattern of net appropriation plays a major role in the North's consumption (Fig. 5). In any given year, the North consumes roughly twice as much labour as it renders, thanks to appropriation through unequal exchange. In 2021, net-appropriated labour comprised 46% of the North's total consumption of labour. Figure 5 also shows that the economies of the global North have become increasingly reliant on low-skilled labour, the vast majority of which is net-appropriated from the global South (71% in 2021).

While the majority of the North's net appropriation of Southern labour is comprised of medium-skilled labour, the net appropriation of high-skilled labour nonetheless constitutes a significant feature of Northern economies. We find that global North economies net-appropriate more high-skilled labour from the global South (52 billion hours in 2021) than they obtain and consume through North–North trade (31 billion hours in 2021).

### The wage value of appropriation through unequal exchange

As previous studies have pointed out[15,21], because wages and prices are an artefact of bargaining power in the world economy (plus the level of commodification, the extent of monopoly concentration, etc.) as well as dynamics of supply and demand, it is not possible to assign a "true" monetary value to labour. The most we can do is to represent labour in terms of the prevailing wages experienced by different agents in the existing capitalist world economy, purely as a point of reference. Previous studies of unequal exchange, including by Samir Amin[9], argue that the net appropriation of hidden labour from the global South should be represented in terms of the prevailing Northern wages; in other words, from the perspective of Northern workers and producers[21]. This is the approach we take here.

First, we established the scale of net-appropriated labour time within each skill level for each year. We then multiplied the net-appropriated labour time by the wages that Northern workers receive for the labour of the same skill level, rendered in the production of traded goods (ignoring goods produced for final domestic consumption). Labour is therefore compared like-for-like: for example, the appropriated quantity of low-skill labour is valued at the Northern wage for low-skill work, the appropriated quantity of high-skill labour is valued at the Northern wage for high-skill work, etc. In this way, we calculated the wage value of appropriated labour in a manner that answers longstanding questions about the degree to which this is affected by the skill level composition of exchange.

Our results show that in 2021 the wage value of labour net-appropriated from the South was worth €16.9 trillion, in constant 2005

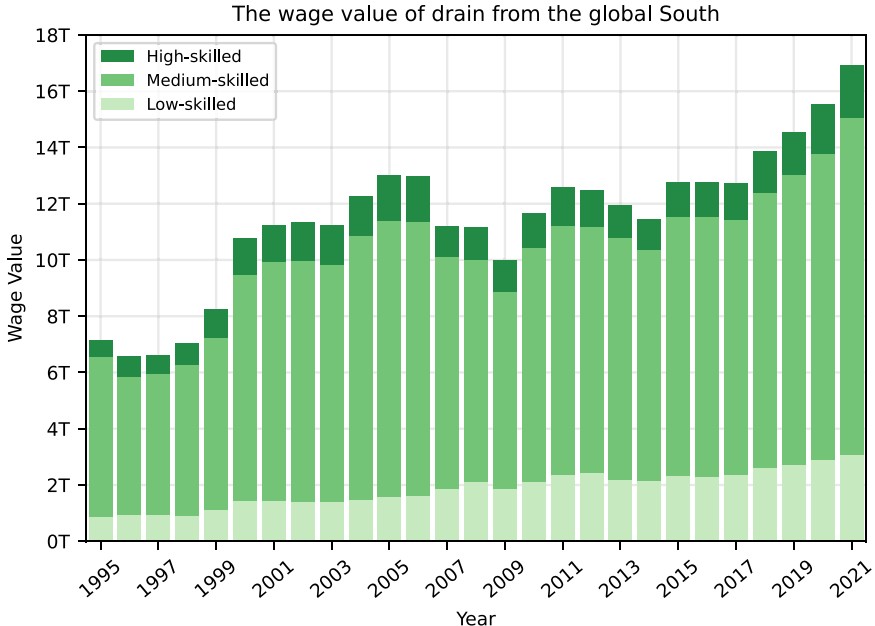

**Fig. 6 | The wage value of labour net-appropriated from the global South in trillion Euros (constant 2005), by skill level, 1995–2021.** Skill levels are shaded from lighter (low-skilled) to darker (high-skilled). Labour here is represented in terms of Northern wages for labour of a given skill level rendered in the production of traded goods.

EUR (Fig. 6). In other words, if Northern workers were to perform the net-appropriated quantity of labour domestically, it would cost €16.9 trillion in terms of wages (see Supplementary Discussion 1 for further interpretation). The time series shows that the wage value of the appropriation has more than doubled since 1995. Large increases occurred during the late 1990s, continuing an upward trajectory that began during the period of neoliberal structural adjustment in the 1980s (see evidence presented in previous work[15]). The appropriation plateaued from the early 2000s to 2015, and since then, has increased further. Over the period 1995–2021, the total wage value of net-appropriated labour sums to €310 trillion.

For robustness, we also calculated the wage value of net-appropriated labour using Northern wages for each skill level in the relevant sector (in other words, we valued the net-appropriated quantity of high-skill labour in the services sector at the Northern wage for high-skill work in the services sector, etc.). The results are only marginally lower, €14.2 trillion in 2021. The weakness of this approach is that it takes for granted large wage inequalities between sectors within the global North, even when correcting for skill, which are due in large part to extremely low wages in agriculture (as low as 2-3 euros per hour; see Supplementary Fig. 2). However, it is nonetheless useful to demonstrate that the large scale of the wage-value of net-appropriated labour cannot be explained by sectoral differences.

### Wage trends

We find that the North–South wage gap is large and has been increasing over time for all skill level categories (Fig. 7). Here again, we assess labour involved in the production of traded goods only. Southern wages are 87–95% less than Northern wages at the same skill level, i.e. for equal work as defined by the ILO. Southern wages are 87% less for high-skill labour, 93% less for medium-skill labour, and 95% less for low-skill labour. The disparity is so extreme that high-skill labour in the global South receives 68% less than low-skill labour in the global North. Stated otherwise, for every hour of work at a given skill level, Northern workers are able to consume 8–19x more of the global product than Southern workers (8x more for high-skill labour, 14x more for medium-skill labour, and 19x more for low-skill labour).

Southern wage gains have not matched Northern wage gains in absolute terms. The average Southern wage has increased from €0.46

to €1.62 per hour (an increase of €1.16), while the average Northern wage has increased from €12.60 to €24.95 per hour (an increase of €12.35). Northern wages have increased 11x more than Southern wages. There is no "catch-up" occurring; on the contrary, it is a pattern of dramatic divergence.

These results indicate that workers in the global South, who receive an average €1.62 per hour, perform the vast majority (90%) of the labour that produces for the global economy, the vast majority (91%) of the labour that produces traded goods, and nearly half (46%) of the labour that supports growth and consumption in the global North (net of trade). The global economy is overwhelmingly characterised by a regime of cheap labour.

These wage gaps are not explained by sectoral differences. Our results show large and growing North–South wage gaps for all skill levels across all of the sectors we analysed. In agriculture, Southern wages are 85–91% lower than Northern wages for any given skill level. In mining, Southern wages are 93–98% lower. In manufacturing, 89–94% lower. In services, 83–90% lower (see Supplementary Fig. 2 for full results).

Despite growing wage gaps, there has been some reduction in the relative inequality between North and South. As of 2021, the average Southern wage was 94% lower than the average Northern wage, a small improvement from 96% in 1995 (Fig. 8). Improvements in the South's position occurred during the period from 2005 to 2015. Improvements have since ceased and, to some extent, declined.

### Labour's share of GDP

We find that, globally, labour received, on average, 51.6% of world GDP during the 5-year period 2017–2021. In other words, only half of all value produced in the world economy (that is represented in prices and included in GDP accounts) is captured by workers in the form of wages. As Fig. 9 shows, this represents a decline from the late 1990s (1995–1999), when labour's share of GDP averaged 54.7%. This suggests that labour's position vis-à-vis capital has deteriorated over the period anlaysed.

Southern workers' share of Southern GDP is notably lower than the global average, at an average of 47.5% during the 2017–2021 period, while Northern workers' share of Northern GDP is higher, at an average of 54.7% during the same period. This indicates that the

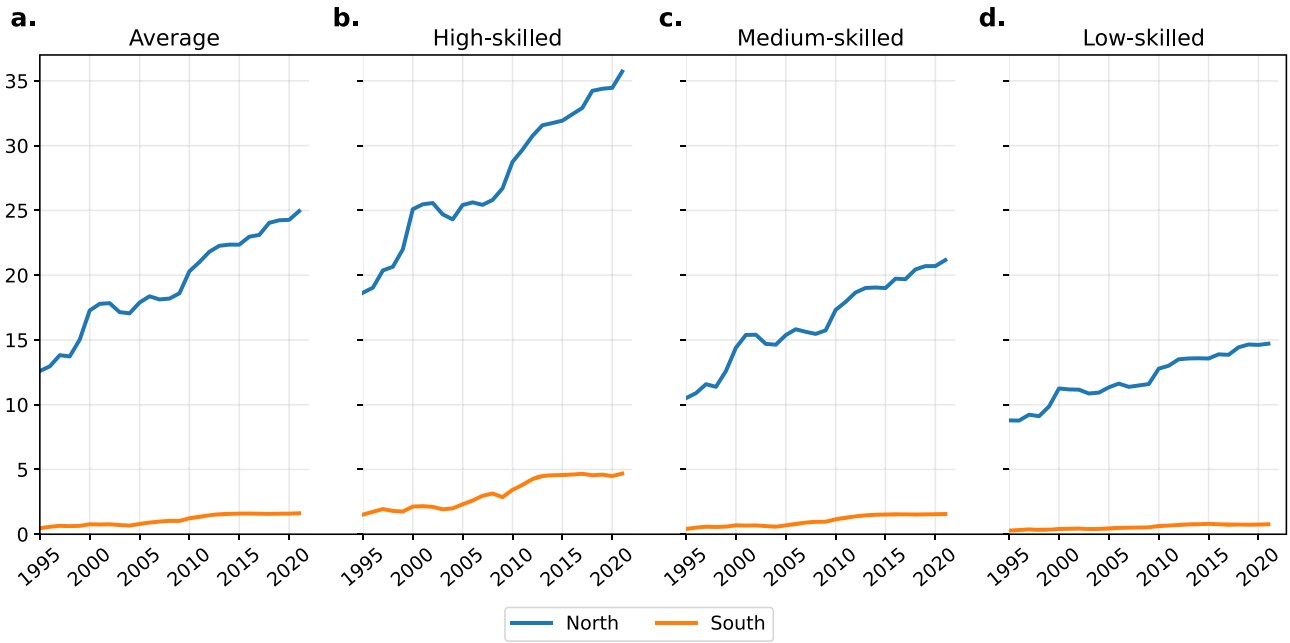

**Fig. 7 | Wage trends in the global North and South by skill level, Euros per hour (constant 2005), 1995–2021.** Global North wages are in blue, global South wages are in orange. Panel **a** shows the average wages of all labour, across all skill levels. Panel **b** shows the wages of high-skilled labour. Panel **c** shows the wages of medium-skilled labour. Panel **d** shows the wages of low-skilled labour.

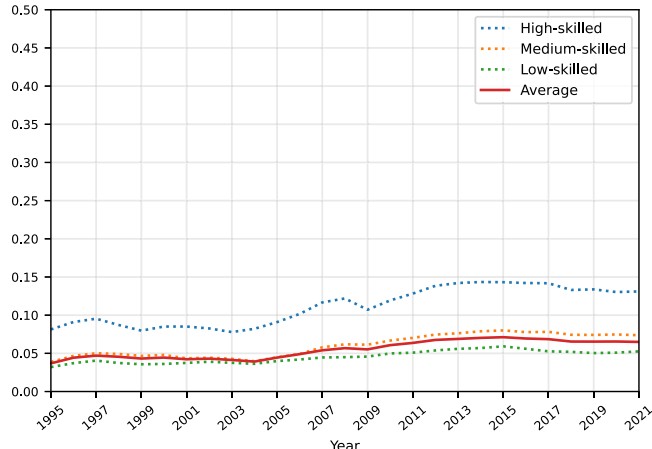

**Fig. 8 | Southern wages as a ratio of Northern wages, by skill level, Euros per hour (constant 2005), 1995–2021.** The figure shows Southern wages for high-skilled labour (blue dotted line), medium-skilled labour (orange dotted line), low-skilled labour (green dotten line), and the average Southern wage (red solid line).

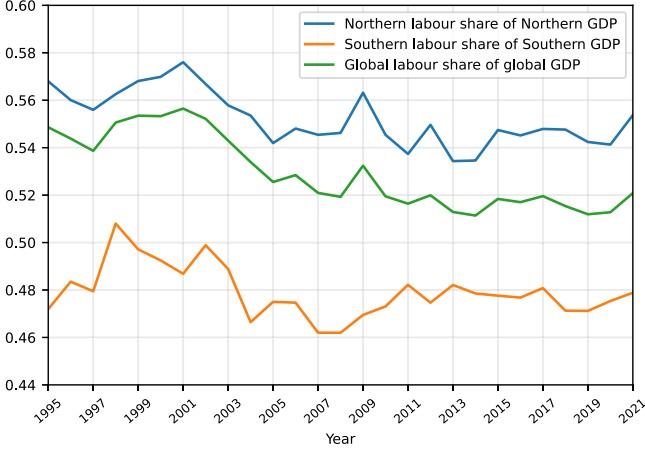

**Fig. 9 | Labour's share of GDP by region (constant 2005 EUR), 1995–2021.** The figure shows the Northern labour share of Northern GDP (blue line), the Southern labour share of Southern GDP (orange line), and the global labour share of global GDP (green line).

Southern working classes are weaker vis-à-vis national capital than the Northern working classes. In both cases, labour's position has deteriorated since the late 1990s: by 1.3 percentage points in the South and 1.6 percentage points in the North.

## Discussion

The results of this study demonstrate the occurrence of large net transfers of embodied labour from the global South to the global North through unequal exchange in international trade and global supply chains, across all skill levels and all sectors, amounting to a total of 826 billion hours in 2021. These results illustrate several important features of the world economy. For one, it is clear that Northern economies rely substantially on net appropriation from the global South. Net-appropriated labour comprises approximately half of the total labour

that provisions goods and services for Northern consumption. In other words, this dynamic doubles the total quantity of labour that is available to the economies of the global North, which sustains their high levels of consumption and wealth, and underpins their economic growth. An extra 826 billion hours of Southern labour is effectively appended to Northern economies as invisible "ghost workers".

Given this dynamic, it is clear that the North's development model cannot be universalised, as it relies on appropriation from elsewhere. Furthermore, it is unlikely that the North's current levels of aggregate consumption could be maintained under fair trade conditions. To maintain current consumption, Northern populations would need to substantially increase their working hours (while also committing substantially more domestic land, materials and energy to production), which would be socially and politically difficult to achieve. It is

plausible that people would prefer instead to forego some kinds of production (for instance, production of goods for elite consumption) or shift to forms of provisioning that require less labour (e.g. public transit instead of private cars).

Our results also indicate that the global South is drained of a large quantity of productive capacity through unequal exchange (9–16% of its total productive capacity, in terms of labour, is drained in any given year). 826 billion hours of labour in 2021 is equivalent to 369 million workers (assuming 2236 h per worker per year, which is the global South average as presented in Table 1). This is more than the total workforce of the United States and the European Union combined. This quantity of labour could be mobilised to produce housing and nutritious food for communities within the global South, or to build and staff hospitals and schools, thus provisioning for local human needs and achieving necessary development objectives; but instead—because of the squeezing of Southern labour and producers, and because of constraints on the ability of Southern states to develop greater economic sovereignty—it is appropriated to produce within global supply chains that service Northern growth, consumption and accumulation.

Unequal exchange is understood to be driven in part by large North–South wage gaps. We find that Southern wages are 87–95% lower than Northern wages for work of equal skill. Our analysis confirms arguments by others (e.g. Ruy Mauro Marini, Samir Amin, Arrighi Emmanuel, etc.) that these wage gaps cannot be explained by sectoral differences, as they prevail across all sectors of the economy: the wages of Southern workers are 83–98% lower for work of equal skill within the same sector. In absolute terms, these wage gaps have increased substantially over time, across all skill levels and all sectors, indicating a steady increase in absolute North–South income inequality, despite some improvement in the South's relative position during the 2005–2015 period. This has occurred even though there has been a steady increase in industrial manufacturing as a share of total production in (and exports from) the global South during the period studied[13,14]. We conclude that while Southern workers contribute the majority (90–91%) of labour that powers the world economy, they receive only 21% of global GDP. The yields of production are disproportionately captured in the global North.

It is interesting to note that our results describing the North's reliance on high-skilled labour from the South confirm statements made by the CEOs of major firms, who have claimed that the necessary quantity of high-skilled engineering labour to support their production is not available in the core states and therefore this production must be performed abroad. Apple's former CEO Steve Jobs noted that his company required 30,000 engineers; as he put it, "You can't find that many in America to hire"[26]. Current CEO Tim Cook has noted that Apple's production relies on large quantities of high-skilled labour in China for advanced engineering, tooling and innovation. "In the US, you could have a meeting of tooling engineers and I'm not sure we could fill the room. In China, you could fill multiple football fields"[25]. High-skilled labour in the global South is paid a fraction of that in the North, despite the fact that Northern firms cannot function without it.

Our study shows that North–South wage gaps and the unequal exchange of labour cannot be explained by skill level or sectoral differences, as both dynamics persist within all skill levels and within all sectors. However, some portion of unequal exchange may be due to productivity differences (i.e. if workers in the South produce less output per hour than workers in the North). Unequal exchange is sometimes conceptualised as a transfer net of productivity differences[9,10]. It is not straightforward to assess this empirically. Standard metrics of productivity measure output in terms of income or GDP per person employed, which are metrics of prices and do not reveal anything about physical output. Such metrics cannot be used for our purposes: they tell us that Southern wages and prices are lower than Northern wages and prices, but this is precisely what needs to be explained[27,28]. Scholarship on unequal exchange holds that prices are an artefact of

power imbalances (between labour and capital, suppliers and lead firms, periphery and core) and, therefore, cannot be taken as an accurate depiction of value. If Southern wages and prices are squeezed, Southern 'productivity' will appear to deteriorate even if there is no change in the actual physical output. To properly assess productivity dynamics would require measuring physical output across comparable industries (see Supplementary Discussion 2 for further discussion).

There are several reasons to believe that physical productivity differences cannot explain the large North–South wage gaps and unequal exchange we observe. First, in the case of export industries (rather than subsistence and non-traded sectors), most production in the South is performed with modern techniques, often with technologies provided by international capital[9,10,14,20]. Workers in these industries have been found to produce as much or more physical output per hour than their Northern counterparts[20,29]. Southern production is also characterised by greater labour intensity, as workers are subject to rigid systems of control designed to maximise output to an extent that would fall foul of labour regulations in the core[30–32]. Second, in cases where Southern export industries do operate with less efficient technologies, productivity differences only account for a small share of the North–South wage gap and the net appropriation of labour time[9,20]. The key fact is that Northern firms choose to use Southern labour not only because wages are cheaper per hour, but because wages are cheaper per unit of physical output[13]. Offshoring occurs precisely because the North–South wage difference is greater than any physical productivity difference. Third, physical productivities can only meaningfully be compared for identical tasks and products. For many industries and product categories, Southern production has no counterpoint in the North, because it cannot or does not occur there (such as in the case of coffee, coltan, smartphones, fast fashion, etc.)[9,10,33]. In such cases, productivities cannot be compared and cannot explain wage inequalities and unequal exchange. Further analysis is available in Supplementary Discussion 2.

It is important to note that, in cases where physical productivity differences do exist, this is often because it is more profitable for capital to use cheaper, more labour-intensive methods than to invest in modern equipment—especially in cases where state investment in technological development has been curtailed by structural adjustment programmes, or where patents prevent affordable access to necessary technologies—precisely because Southern wages are maintained at artificially low levels[34,35]. This arrangement benefits Northern consumers with cheaper goods and benefits Northern capital with an increased surplus. In such cases, the use of labour-intensive methods facilitates value transfer and should be understood as constituting unequal exchange. Under these conditions, the South is compelled to allocate more labour to production for international trade than would be required if technology was deployed more rationally and fairly, thus draining—and wasting—a crucial productive capacity that could otherwise be allocated toward producing goods and services necessary for local well-being and development (see Supplementary Discussion 2).

This study has assessed flows of embodied labour between the core and periphery. Such an approach is clearly useful for broad world-system analysis, but it may obscure other important dynamics. Future research may find it useful to explore a more detailed regional breakdown, to assess how the periphery and semi-periphery (or low-income and middle-income countries) are differently affected by unequal exchange. It would also be worth assessing class dynamics, and inequalities within countries. It should be understood that unequal exchange is ultimately driven by the corporations and investors that control supply chains, and the states that determine the rules of international trade and finance, not by workers or consumers. Patterns of unequal exchange may also operate within countries, through the exploitation of 'internal peripheries'[36] (such as extremely low-wage agricultural labour in the core, as visible in Supplementary Fig. 2). Finally, our analysis does not extend to gender dynamics, including

uncompensated household labour or social reproduction, which is also constitutive of unequal exchange and should be explored in future EEMRIO research[37,38].

The results of this study suggest that the persistence of global poverty and underdevelopment is, in large part, an effect of appropriation through unequal exchange, which is, in turn, an effect of wage suppression or income deflation in the periphery. People in the global South have their consumption curtailed such that labour, resources and goods are more readily available for appropriation by Northern states and firms. This dynamic also helps us understand persistent inequality between the core and periphery. Under conditions of unequal exchange—where production in poorer countries is appropriated for consumption in richer countries—convergence is not feasible to achieve. Development and poverty eradication, and any plausible trajectory for reducing global inequality, requires a shift in the balance of power between North and South, such that the latter is able to reclaim its productive capacities to meet human needs.

Toward this end, international wage floors and minimum resource prices could help reduce price inequalities and limit value transfers. Ending unequal exchange will also require ending structural adjustment conditions on finance, and democratising the institutions of global economic governance, so that global South governments are free to use industrial, fiscal and monetary policy to pursue sovereign development and reduce their dependency on Northern capital. Such reforms are unlikely to be handed down from above, however. It will require a political struggle for national self-determination and economic sovereignty similar in scope to the anti-colonial movement of the 20th century.

## Methods

### Deriving data from EXIOBASE

Input-output (IO) tables record the exchange of goods and services between various sectors within an economy, incorporating intermediate demand ($Z$), final demand ($y$) and value-added ($v$)[39]. Complementing the monetary IO tables, extension tables record non-monetary flows connected to economic activities such as raw material extraction, energy usage, land use and labour requirements. By blending data on the economic structure of supply chains from multi-regional Input-output (MRIO) tables with information on the environmental and socioeconomic pressures that emerge throughout the supply chain, environmentally extended multi-regional input-output (EEMRIO) analysis allows for the assessment of both socioeconomic and environmental impacts throughout the supply chain network that ultimately reaches the final consumer, a phenomenon known as 'footprinting'[40]. For a more thorough introduction to EEMRIO, see ref. [41].

The EEMRIO method employs an MRIO table that records inter-regional trade between sectors across countries. If $y$ is a matrix of final demand, the total output of sector $i$ ($\mathbf{x}_i$) would be given by the sum of row $i$, where $\mathbf{z}_{ij}$ is the intermediate sales from sector $i$ to sector $j$, and $\mathbf{y}_{ic}$ is the final demand for products of the sector $i$ by country $c$ (ref. [42]):

$$\mathbf{x}_i = \mathbf{z}_{i1} + \mathbf{z}_{i2} + \ldots + \mathbf{z}_{ij} + \mathbf{y}_{i1} + \mathbf{y}_{i2} + \ldots + \mathbf{y}_{ic}$$

The demand-driven IO framework can be calculated using the equation:

$$X = (I - A)^{-1}\mathbf{y} = L\mathbf{y}$$

where $A$ is the direct requirements matrix. Here, the element $a_{ij}$ gives the direct inputs that sector $j$ requires from the sector $i$ ($\mathbf{z}_{ij}$) in order to produce a unit of output $\mathbf{x}_j$ (ref. [42]):

$$a_{ij} = \frac{\mathbf{z}_{ij}}{\mathbf{x}_j}$$

$L = (I - A)^{-1}$ is known as the 'Leontief inverse,' whose elements quantify the total upstream direct and indirect inputs from sector $i$ required to produce one unit of industry output $j$ for final demand. $L$ approximates the potentially infinite chain of production required to produce a unit of final product[43]; for example, an iPad sold in the United States might be assembled in China using components manufactured in South Korea, which in turn requires subcomponents produced in the Philippines from raw materials extracted and processed in other countries and sectors, etc.[19]. Multiplying $L$ by final demand $\mathbf{y}$ yields the total upstream inputs required to produce the goods and services consumed by households, governments, and others, $X$, in monetary units.

Environmental or socioeconomic coefficients, which determine pressures or impacts, are stored in the intensity vector $\mathbf{q}$ representing the pressure or impact per unit of industries' total output, such as the hours of labour required to produce a unit of output. For our purposes, we want to calculate the labour from sector $i$ of producer country $p$ embodied in final demand of consumer country $c$. This simply requires multiplying the labour requirements per unit output of sector $i$ in country $p$ (stored in vector $\mathbf{q}$) with the total output from that same country and sector required to meet final demand in consumer country $c$ (stored in matrix $X$). If $\mathbf{y}$ is a single-column vector of final demand, the flow matrix $F$ can be expressed in matrix notation as:

$$F = \hat{\mathbf{q}}L\hat{\mathbf{y}}$$

where a 'hat' (^) represents a diagonalized vector; it shows the labour inputs by sector and region required to satisfy each product category of final demand $\mathbf{y}$[44].

We applied EEMRIO analysis using data from the MRIO database EXIOBASE v.3.8.1[45]. EXIOBASE 3 presents a comprehensive and updated series of EEMRIO tables for 200 product sectors, covering a span of time from 1995 up to a recent year. The data comprises 44 individual countries, consisting of 28 European Union member states as well as 16 other major economies, in addition to five regional aggregations that together cover the rest of the world. To obtain consumption-based labour flows between the selected categories, we used the socioeconomic coefficients for the labour accounts, including labour time (which is represented in hours) and compensation (which is represented in constant 2005 euros) for each skill level. We examined the period from 1995, which is the earliest year available in EXIOBASE v.3.8.1, to 2021.

One key limitation of EXIOBASE is the use of now-casting since 2011[46]. However, given we have no reason to expect large changes in labour efficiency or wage trends in the latter period, nor sudden changes in the structure of the world economy, we believe it is reasonable to use the now-casted data for our purposes.

### Regional aggregations

Our analysis here focuses on flows between the core and periphery of the world system, using the country classifications set out in Supplementary Table 1. As a proxy for the core, or the global North, we used the IMF's list of 'advanced economies' as a guide and created the closest possible approximation of this list given the countries available in EXIOBASE. The category includes USA, United Kingdom, Canada, Australia, Norway, Austria, Belgium, Germany, Denmark, France, Luxembourg, Netherlands, Finland, Sweden, Switzerland, Japan, South Korea, Estonia, Spain, Greece, Ireland, Italy, Latvia, Malta, Portugal, Slovenia, Slovakia, Taiwan, Cyprus and the Czech Republic.

The periphery, or global South, includes all other countries (i.e. the IMF's 'emerging and developing' countries). In EXIOBASE, several of the IMF's 'advanced economies' (Singapore, San Marino, Iceland, Israel, Liechtenstein, Macao SAR, Hong Kong, Puerto Rico, Monaco, Bermuda, Andorra and New Zealand) are aggregated into regions, such as 'Rest of Europe', 'Rest of Asia', etc. We were, therefore, compelled to include these countries in our 'global South' category. The

**Table 2 | Description of ILO skill levels, as used in EXIOBASE**

| Skill type | Occupations | Education attainment levels |
|---|---|---|
| Low-skilled | -Elementary occupations | -Less than primary education<br>-Primary education<br>-Lower secondary education |
| Medium-skilled | -Plant and machine operators, and assemblers<br>-Craft and related trades workers<br>-Skilled agricultural, forestry and fishery workers<br>-Services and sales workers<br>-Clerical support workers | -Upper secondary education<br>-Post-secondary non-tertiary education |
| High-skilled | -Technicians and associate profes-sionals<br>-Professionals<br>-Managers | -Short-cycle tertiary educa-tion<br>-Bachelor's or equivalent level<br>-Masters or equivalent level<br>-Doctoral or equivalent level |

consequence is that our results for wages in the global South are likely to be slightly higher, and our results for labour drain from the global South slightly lower, than what would otherwise be the case if we were able to match the IMF categories more accurately. These are small countries, however, and the effect is likely to be minimal.

### Skill levels

In EXIOBASE, skill level categories are defined according to the International Labour Organisation's classification scheme for low-skilled, medium-skilled and high-skilled labour. Table 2 summarises the ILO categories by occupation and educational attainment, based on the International Standard Classification of Occupations (ISCO). The ILO offers extensive descriptions of each occupation, available in further documentation[47]. For instance, 'Elementary occupations' involve the performance of simple tasks which may require the use of hand-held tools and physical effort, such as cleaning, restocking supplies, performing basic maintenance in buildings, performing manual labour in farming, fishing, mining, manufacturing, construction, etc.

### Sectors

We divided EXIOBASE production data into five sectors (agriculture, manufacturing, mining, services, and 'other'), based on recommended sectoral aggregation of product tables (available at https://ntnu.app.box.com/v/EXIOBASEconcordances/file/282981183372). The 'other' category aggregates the electricity and utilities, construction, and transport sectors. Full details on the sectoral composition are available in Supplementary Data 1.

### Monetary units

EXIOBASE reports monetary units in constant 2005 euros, which control for inflation. These units are expressed in MER (with local currencies converted at the market exchange rate) rather than PPPs. PPPs are useful for understanding people's purchasing power over goods within a national economy. However, MER are more appropriate for understanding purchasing power over goods in the world market, which is the purpose of this paper. It is well established in political economy that PPP methods are not appropriate for measuring relative international wealth and power, or command over the global product[12,48,49–51]. According to Arrighi, "While PPP data allow for a more adequate description of trends in material consumption, FX-based data are a better measure of differences in the relative level of income/wealth among residents of different countries in the global economy. Wealth in a global economy is the command that people have over one another's goods and services on the world market. PPP-adjusted data actually obscure what we seek to measure"[48]. As Ricci writes, for these reasons "It is preferable to use… market exchange rates to compare income purchasing power in the global economy"[12].

### Overview of calculations

We obtained data on labour embodied in traded goods and services flowing from North to South, from South to North, between Southern countries and between Northern countries, for the three skill levels and the five sector aggregates indicated above, along with the labour compensation paid against these flows. We also obtained data on labour by skill level and sector involved in domestic production and consumption within both the global North and the global South (i.e. labour involved in the production of non-traded goods), along with compensation paid. To calculate the Northern net appropriation of labour, we subtracted Northern flows to the South from Southern flows to the North. To calculate total Northern consumption, we summed Southern flows to the North together with flows of traded goods within the North and production and consumption of non-traded goods within the North. Wages (euros/hour) for each region are calculated as the total compensation (in euros) the region receives for its exports of labour divided by its total exports of labour (in hours). To calculate labour hours per worker we divided total labour hours rendered in each region by the total number of workers employed, also obtained from EXIOBASE.

### Reporting summary

Further information on research design is available in the Nature Portfolio Reporting Summary linked to this article.

## Data availability

All data were derived from EXIOBASE, as described in the Methods. EXIOBASE can be accessed at https://www.exiobase.eu/.

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

## Acknowledgements

We are grateful to Dylan Sullivan for contributing insights and feedback on the draft manuscript and Supplementary Discussions, particularly related to sections on productivity. J.H. and M.H.L. acknowledge support from the European Research Council (ERC-2022-SYG reference number 101071647) and the María de Maeztu Unit of Excellence (CEX2019–374 000940-M) grant from the Spanish Ministry of Science and Innovation. F.B. acknowledges funding from the Swedish Research Council Formas (2021-01006) and the Marianne and Marcus Wallenberg Foundation (MMW2023.0023).

## Author contributions

JH: conceptualization, supervision, project administration, resources, methodology, formal analysis, writing—original draft, writing—review and editing. MHL: conceptualization, software, methodology, validation, data curation, investigation, formal analysis, writing—review and editing. JH and MHL conceptualized and completed the initial submission, FB contributed following the first review. FB: software, methodology, validation, data curation, investigation, formal analysis, writing—review and editing.

## Funding

## Competing interests

The authors declare no competing interests.
