## [Peer Review File · Nature Communications]

Unequal exchange of labour in the world economyReviewers' Comments:

Reviewer #1:

Remarks to the Author:

Review Report for Manuscript NCOMMS-23-36667

Title: Unequal exchange of labour in the world economy

by Andrea Ricci

The paper empirically analyses the contribution of North and South labour to world total and traded production. Unequal exchange results from the difference between the labour input supplied and the value added appropriated by each region of the world economy. The disaggregation of output into 163 industries and of labour input into three categories allows for productivity differences across industries and labour skills. The results show that differences in industrial specialisation and labour skills play a negligible role in explaining North-South inequalities in global income distribution. The degree of disaggregation of the analysis, never before undertaken in such detail, makes a substantial contribution to the empirical literature on unequal exchange, providing new evidence that confirms the main arguments of the theory.

However, there are some theoretical and methodological issues that require further clarification by the author(s).

1) The analysis accounts for intersectoral and inter-skill productivity differentials, but not for intra-sectoral and intra-skill differentials between countries. The key assumption underlying the calculation of labour input is that productivity within industries and skill levels is identical across countries. The Discussion addresses this crucial assumption by distinguishing between monetary and real productivity (lines 325-356). On the former, the only comment concerns the assertion that "In political economy, the prices of labour and goods are understood to be an artefact of power imbalances" (lines 328-9). In fact, it would be more accurate to say that power imbalances determine the deviation of the market price from the cost of production plus average profit for prices and subsistence wages for workers' remuneration.

The discussion on real productivity is more problematic (lines 342-356). As Marx noted, the quantity of physical goods produced per unit of labour time is the result of two factors: labour intensity and labour productivity. Labour intensity is a positive function of the worker's effort per unit of time. Average labour intensity determines the unit of labour time in which labour input in the economy is expressed. As the author(s) rightly point out, there is every reason to believe that workers in the South put as much, if not more, effort into their work as workers in the North. As a result, labour intensity could be considered equivalent in both regions. In contrast, labour productivity is a positive function of physical capital per worker, which is higher in northern countries than in southern countries, as shown by UNCTAD's revealed factor intensity data (see McLaren et al. 2018). The difference is due to a higher stock of accumulated physical capital and to a better level of technology in the North. (On technology, the text's reference to the WEF's Technology Adoption Index in line 347 is unclear). This means that, given the same labour intensity, the physical product per worker is higher in the North than in the South for the same industry and skill level. As a consequence, international comparisons need to define a unit of labour that is homogeneous in terms of both labour intensity and productivity (see Ricci 2019). For countries with higher productivity, the input of homogeneous labour will be a multiple of actual labour, and vice versa for countries with lower productivity. The non-homogenisation of labour in the world economy implies that the paper overestimates the unequal exchange in terms of labour units.

2) The theoretical framework underlying the empirical analysis is based on a Leontief input-output model, which assumes that labour is the only original input that determines net monetary output or value added. Material inputs (raw materials, intermediate goods) contribute to net physical product but not to net monetary product, either because they are free natural resources or because inter-industry purchases and sales offset each other in aggregate. Value added is distributed as income in

the form of wages and profits or gross capital income.

The paper first determines unequal exchange in terms of labour input and then converts it into money by multiplying labour input by Northern monetary wages. Following this procedure, the author(s) attribute the unequal exchange to the difference in monetary wages between the North and the South. However, this method is open to question.

The hours of labour appropriated by the North through unequal exchange correspond to a share of the South's net physical product, which can be expressed in monetary terms as value added by applying market prices. The corresponding monetary value of the labour input appropriated by the North should therefore be calculated in terms of the monetary value of the net physical product, i.e. in terms of value added, which includes both wages and profits. As a result, it does not seem correct to argue that unequal exchange is due to higher money wages in the North. It could just as well be argued that it is due to higher profits. Indirect empirical support is provided in the paper by the fact that it is easy to calculate that profits per unit of labour are higher in the North than in the South. In fact, unequal exchange results from the North's appropriation of a larger share of global value added (wages and profits) than the corresponding net physical product of its labour input. The paper provides sophisticated empirical evidence of this phenomenon, but no answers to the underlying reasons. The monetary conversion of labour input through wages implies that the paper underestimates the unequal exchange in terms of monetary units.

In conclusion, I wish the paper to enter the literature because it adds to the empirical evidence on the unequal exchange, in particular by taking into account the different skill levels of the labour force. My hope is that these comments will be helpful to the author(s) in the clarification of some theoretical and methodological issues in the revision process of the paper.

REVIEW REFERENCES

- Mc Laren A., Saygili M. and Shirotori M. (2018), "Revealed factor intensity of products: Insights from a new database", UNCTAD Research Paper No. 16.
- Ricci A. (2019), "Unequal exchange in the age of globalization", *Review of Radical Political Economics* 51(2): 225-45.

Reviewer #2:

Remarks to the Author:

The research presented in this manuscript addresses "the process associated with the labour time involved in North-South trade" (57-58) and the role of level of skill associated with the labor mainly embodied in exports in accounting for wage differences (if I got this right – the wording around the research gaps is not terribly clear and could be rendered more precise). It's an interesting topic and potentially very productive empirical work, but the communication in this manuscript is always clear or complete and could be improved significantly.

Introduction

Throughout the introduction, it's not quite clear to me whether the argument is one of continuity of colonial relations today (this is stated) or of a shift and increased offshoring on the part of actors in the Global North (this is also claimed). It might be useful to make explicit whether both are thought to be occurring, but then also when the reshoring happened in between.

35-37: For the claim that certain actors "cheapen" labor and what this means, it would be helpful to have an argument and a reference or two.

50: I am not familiar with the term "monopsony" and would appreciate an explanation.

Methods

The methods section is not terribly precise in terms of wording. I would generally recommend rigorous language-editing for this manuscript. I have included some questions below that may have more to do with language than with content.

I'm not convinced that this article should include, in its main body, a generic introduction to IOA. This

might be moved to an annex, or the reader might be directed to other publications. This would free up space for a description of the method that was actually used for this analysis. What indicators were used? In what units? What calculations were performed? Were all data points consistent across the entire time series? How was inflation addressed? What about exchange rates from local currencies to the currency of the model (€)?

An annex detailing the countries (or regions) considered to represent the Global North and South for the purposes of this research would be helpful, unless "the wealthy economies of Europe" means the 28 EU member states (then please state that explicitly)? Is my understanding correct that the United Kingdom, Switzerland, and Norway are not included in this analysis? Maybe a clarifying list would be really helpful.

What are the implications for the analysis of data on (some) of the countries in the Global North being available in national detail and the countries of the Global South only as part of regional groupings?

80: What are supply chain networks and how are they captured in EEMRIO models?

81: In what sense does the 'supply chain network' reach the final consumer?

Table 1: Could you provide more detail on what 'elementary occupations' and professionals are? I am not familiar with the terms and they are not self-explanatory.

Results

Due to the lack of detail in the description of the method and the data (in the labor extension of the MRIO model, in particular), it is quite difficult for me to interpret the results. In order to interpret the results, it would be additionally helpful to have data on the population of the Global North and South (as defined for this study) and their age distribution.

This is also actually a methods question, but the results consistently refer to traded goods. Does this include services?

Again, probably methods-related: There is a jump in Global South low-skilled labor between 2006 and 2007 that seems to be 'compensated' by a drop in medium-skilled labor, making this look like a change in classification. Can you confirm or otherwise comment?

Table 2: What's the reasoning for normalizing labor hours by the total population rather than by working-age population? Why do the changes observed mean that more population is entering the workforce? Couldn't they also mean that the same (or fewer people) are working more? What indicator provides the number of workers per country?

Figure 2: It might be informative to show these results for the Global North as well? So, to show net-flows of embodied labor for each of the two regions? Because, as far as I understand, the export flows in Figure 2 could also be flows from the Global South to the Global South, right? I see that this data is presented in Figures 3 and 4. It might be useful to make the relation, especially between Figures 2 and 4 clear: the darker gray segments of the bars in 4 correspond to the net-exports in 2?

Figure 3 (methods-related): how was net-appropriation calculated? As labor embodied in exports minus labor embodied in imports? How were re-exports accounted for?

Methods-related: Did you account for the relative purchasing-power of wages at all? This partially relates to currency conversion, which is not discussed in the methods chapter! I understand that your argument goes more strongly into the direction of 'these are the wages that would have had to be paid in the North if workers there had done the work themselves'; it might be useful to make this argument (and what it does and does not capture) more explicit.

253: Could you provide an explanation as to why the focus on traded goods here 'maximizes compatibility'? Isn't part of the argument around the world-systems that the periphery actually produces distinctly different goods for export than do the countries of the core?

251-260: In terms of the growth rate (not in absolute terms, as you point out), wages in the South grew much more dramatically (+250%) than in the North (98%), which has to do with the low level, of course, but might be worth watching out for in the wording.

Figure 6: I'm still not clear about why purchasing power is not relevant here, both between the North and the South and within each of the regions over time. Do 2005 Euros have the same purchasing power in 2020 as they did in 2005? Doesn't this matter for our interpretation of wage increases?

Methods-related: Could you please also describe, in the methods chapter, the particular approach to GDP calculation used here? I'm more used to seeing the relative power of labor discussed in terms of

income from labor vs income from capital/wealth and it would be great to know what the remaining 50% or so of GDP consist of in this case. It would be the value added approach? So that in the Global South, labor is somewhat more strongly exploited in a Marxian sense than in the North?

Discussion

The argument in the paper would be more compelling if the discussion focused on the results presented rather than bringing in a new topic (wage inequality) that is not really covered in the results. Similar concern applies to the discussion of the role of technology and the MER vs. PPP aspect, which – you know from my questions – I have been wondering about, but which is not raised in the methods section.

General comments

The manuscript will require language-editing before publication.

Reviewer #3:

Remarks to the Author:

While overall the quantitative work is compelling, the manuscript makes large errors when it comes to the intellectual history of unequal exchange, these should be corrected before it can be considered suitable for publication.

- 1) The author has the right to focus on unequal exchange but should make clear that other dynamics obtained in the historical constitution of cores and peripheries including inter alia settler colonial land alienation/genocide during the mercantilist and early imperial period; and violent political engineering (coup d'état etc) and sanctions during the later colonial and neo-colonial periods; Bagchi and Kadri are essential here
- 2) Theorists of unequal exchange from Marini to Amin to Alrghiri Emmanuel to all work on the North African region make clear that the issue is not "raw" commodities versus industry; this is clear in the preface fo Emmanuel but also later work on export-oriented industrialization; and the even later work of e.g. Intan Suwandi and John Smith; the authors' discussion of this is muddled
- 3) There is no gender analysis; of course uncompensated household labor (or broader social reproduction) is not visible in these statistics but that must be noted; see for example the work of Naidu and Ossome on the agrarian question of gendered labor;
- 4) On p 11 line 324 this would accordingly call for revising the judgment: agricultural labor is not adequately accounted for in the above statistics
- 5) The line about internal peripheries seems a bit throw-away; the dependency analysis is premised on labor reservoirs which themselves are implicitly 'internal' peripheries

RESPONSE TO REVIEWERS

We are grateful for the reviewers' comments, which have helped us improve the paper considerably. We opted to advance our study further by assessing labour flows and wages not only across skill levels but also across sectors. The results of this exercise demonstrate that large net South-North flows of labour and large South-North wage inequalities persist across all skill levels in each sector, thus confirming that these dynamics are not due to differences in the sectoral composition of production and trade. We believe these results strengthen the paper considerably, and advance the literature in new ways.

We have also added Extended Data figures to represent the full range of our results, as well as a Supplementary Information file with tables detailing the regional and sectoral aggregations as well as extended discussions of the results.

Below we respond to each of the reviewers' points:

REVIEWER 1	RESPONSES
The analysis accounts for intersectoral and inter-skill productivity differentials, but not for intra-sectoral and intra-skill differentials between countries. The key assumption underlying the calculation of labour input is that productivity within industries and skill levels is identical across countries.	→ Thank you for this comment. Our results for trade in embodied labour are derived directly from EXIOBASE and we have not made any assumptions about productivity. We understand that unequal exchange of labour is often defined as net of productivity differences. However, it is not possible for us to assess the extent to which the net flows of labour represented in our results are due to productivity differences. We have included new text to address this question extensively in the Discussion as well as in a Supplementary Discussion file. With respect to note about sectors, we have addressed this matter extensively in the revised text, where we have now added full results of unequal exchange by sector.
The Discussion addresses this crucial assumption by distinguishing between monetary and real productivity (lines 325-356). On the former, the only comment concerns the assertion that "In political economy, the prices of labour and goods are understood to be an artefact of power imbalances" (lines 328-9). In fact, it would be more accurate to say that power imbalances determine the deviation of the market price from the cost of production plus average profit for prices and subsistence wages for workers' remuneration.	→ Thank you for this comment. We understand the point here to be that it is not only disparities in the cost of labour that drive unequal exchange, but also in the scale of profits (which together comprise the total price of traded goods). Squeezing the profits of Southern producers (or increasing the profits of Northern producers) may drive UE as much as squeezing labour. We have edited the manuscript to mention this dynamic in several places.
The discussion on real productivity is more problematic (lines 342-356). As Marx noted, the quantity of physical goods produced per unit of labour time is the result of two factors: labour intensity and labour productivity. Labour intensity is a positive function of the worker's effort per unit of	→ Thank you for the useful comments. Again, our calculations of unequal exchange of labour time are derived directly from EXIOBASE, without any assumptions about labour productivity on our part. We should note however that we are using the term "unequal exchange" here in its absolute sense, as

time. Average labour intensity determines the unit of labour time in which labour input in the economy is expressed. As the author(s) rightly point out, there is every reason to believe that workers in the South put as much, if not more, effort into their work as workers in the North. As a result, labour intensity could be considered equivalent in both regions. In contrast, labour productivity is a positive function of physical capital per worker, which is higher in northern countries than in southern countries, as shown by UNCTAD's revealed factor intensity data (see McLaren et al. 2018). The difference is due to a higher stock of accumulated physical capital and to a better level of technology in the North. (On technology, the text's reference to the WEF's Technology Adoption Index in line 347 is unclear). This means that, given the same labour intensity, the physical product per worker is higher in the North than in the South for the same industry and skill level. As a consequence, international comparisons need to define a unit of labour that is homogeneous in terms of both labour intensity and productivity (see Ricci 2019). For countries with higher productivity, the input of homogeneous labour will be a multiple of actual labour, and vice versa for countries with lower productivity. The non-homogenisation of labour in the world economy implies that the paper **overestimates** the unequal exchange in terms of labour units.

more labour exchanged for less, without attempting to quantify the share of this unequal exchange that may be explained by productivity differences.

We agree that labour intensity should be distinguished from labour productivity, and we have now edited the text accordingly.

Our study does not attempt to correct for productivity differences between North and South because we are not aware of any method that would allow us to do this with any accuracy. We cannot use the McLaren et al data on physical capital per worker for this purpose, because this applies to *total* production when what matters for our purposes is production for export, and because we do not know the relative contribution of each Southern country to total Southern exports (particularly given the regional aggregations in EXIOBASE), and because this does not help when it comes to forms of Southern production that do not have a counterpart in the North.

Also, we note that in many cases foreign capital chooses to use less efficient or more labour-intensive production methods because they are *more profitable* than investing in physical capital, and this facilitates rather than mitigates the transfer of surplus value.

We also note that productivity differences that arise from differential endowment of producer technologies (and physical capital per worker) should be understood as an effect of imperialist dynamics that have prevented technological development in the South; therefore if Southern labour inputs to exported goods are higher than they would otherwise need to be if their technological development had not been arrested, this should be understood as a drain of labour that could otherwise be put to other uses.

Given these complications, we feel the best we can do is rewrite the discussion section to explain why it is unlikely that productivity differences explain the full extent of the net transfers visible in our results, and include in the SI a discussion of the limitations we face in assessing productivity differences. We have done both.

Note: the WEF's technological adaptation index is described in the cited source.

2) The theoretical framework underlying the empirical analysis is based on a Leontief input-output model, which assumes that labour is the only original input that determines net monetary output or value added. Material inputs (raw materials,

→ Thank you for these points. We agree that the value transfer should be understood in terms of value added, which includes both wages and profits, rather than only in terms of the wage value of production. We have edited the text to clarify that

intermediate goods) contribute to net physical product but not to net monetary product, either because they are free natural resources or because inter-industry purchases and sales offset each other in aggregate. Value added is distributed as income in the form of wages and profits or gross capital income. The paper first determines unequal exchange in terms of labour input and then converts it into money by multiplying labour input by Northern monetary wages. Following this procedure, the author(s) attribute the unequal exchange to the difference in monetary wages between the North and the South. However, this method is open to question. The hours of labour appropriated by the North through unequal exchange correspond to a share of the South's net physical product, which can be expressed in monetary terms as value added by applying market prices. The corresponding monetary value of the labour input appropriated by the North should therefore be calculated in terms of the monetary value of the net physical product, i.e. in terms of value added, which includes both wages and profits.	the monetary estimates we give correspond only to the wage-value of net-appropriated labour, rather than total value transfer through unequal exchange. This is an important distinction. The approach you have suggested is interesting (i.e., to treat the hours of labour appropriated by the North through unequal exchange as corresponding to a share of the South's net physical product, which can be expressed in monetary terms as value added by applying market prices). The problem we see with this approach is that the monetary value of the South's net physical product is not an accurate representation of that product, precisely because the prices of Southern labour, resources and output are cheapened relative to Northern prices. We could take the North's net-appropriation of Southern labour as a share of the South's total labour involved in all production (the result is 9-16% in any given year). But applying this to the Southern GDP and using the result to represent the monetary value of the net appropriated labour would be to accept that Southern prices are "accurate". In other words, it would be to accept that GDP per labour hour is an accurate representation of productivity, while the literature on unequal exchange, which we cite in the paper, demonstrates it is not. We have included a brief discussion in the Annex with a calculation of total value transfer, not just the wage-value of net-appropriated labour (basically, a counterfactual scenario where the South had been allowed to invest in technological development), while also recognizing that this can only be understood as a thought experiment.
As a result, it does not seem correct to argue that unequal exchange is due to higher money wages in the North. It could just as well be argued that it is due to higher profits. Indirect empirical support is provided in the paper by the fact that it is easy to calculate that profits per unit of labour are higher in the North than in the South. In fact, unequal exchange results from the North's appropriation of a larger share of global value added (wages and profits) than the corresponding net physical product of its labour input. The paper provides sophisticated empirical evidence of this phenomenon, but no answers to the underlying reasons. The monetary conversion of labour input through wages implies that the paper underestimates the unequal exchange in terms of monetary units.	→ Yes, we agree that the disparity in wages is not the only driver of unequal exchange, the disparity in profits is also at play. As indicated above, we have edited the text to mention this, including observing that "the yields of production are disproportionately captured in the global North." We would like to ask for further clarification of what you have in mind when you say "Indirect empirical support is provided in the paper by the fact that it is easy to calculate that profits per unit of labour are higher in the North than in the South." We are aware that the results of the paper indicate that the North's profits per unit of its labour input are higher than in the South, because the North's consumption of value from labour is much greater than its production of value through labour. The value net-appropriated through unequal exchange in trade is acquired gratis, without any labour input from the

	North. However, if you are referring to some other dimension please let us know.
REVIEWER 2	RESPONSES
The research presented in this manuscript addresses “the process associated with the labour time involved in North-South trade” (57-58) and the role of level of skill associated with the labor mainly embodied in exports in accounting for wage differences (if I got this right – the wording around the research gaps is not terribly clear and could be rendered more precise). It’s an interesting topic and potentially very productive empirical work, but the communication in this manuscript is [not] always clear or complete and could be improved significantly.	→ Thank you for these comments. We have revised the discussion of the research gaps – and the rest of the text – in a way that we hope improves clarity.
Throughout the introduction, it’s not quite clear to me whether the argument is one of continuity of colonial relations today (this is stated) or of a shift and increased offshoring on the part of actors in the Global North (this is also claimed). It might be useful to make explicit whether both are thought to be occurring, but then also when the reshoring happened in between.	→ Thank you for your observation. We have revised the introduction so that this should no longer be a problem, however we want to avoid beginning with a too-lengthy discussion of economic history as this as it has been discussed extensively in the broader literature on unequal exchange, which we cite, and would be cumbersome to reproduce here. To answer your question, the core economies have always relied on an appropriation from the periphery. During the colonial period this was obtained directly, as tribute (as colonizers controlled the output and exports of the colonized territories), whereas in the post-colonial period it is obtained through unequal exchange in international trade and global supply chains. While unequal exchange was comparatively low during the 1960s/70s – during the period of national sovereignty and developmentalism in the global South – it increased dramatically during the 80s/90s under structural adjustment. We have tried to make this clearer in the introduction.
35-37: For the claim that certain actors “cheapen” labor and what this means, it would be helpful to have an argument and a reference or two.	→ We have added citations. In the paragraph on structural adjustment we have added details about cuts to public employment and removal of labour standards etc, also with references.
50: I am not familiar with the term “monopsony” and would appreciate and explanation.	→ Monopsony refers to a situation when there is a single buyer that can dictate prices to suppliers. For instance, McDonalds has significant power to dictate prices to suppliers of potatoes and beef, Apple can dictate prices to Foxconn and other suppliers, etc. However, we have decided to remove the term and replace with less technical language to avoid confusion.
The methods section is not terribly precise in terms of wording, but this may have more to do with the authors’ English proficiency and less with a lack of methodological clarity. I would generally	→ Thank you. We have revised the Methods section accordingly.

recommend rigorous language-editing for this manuscript.	
I'm not convinced that this article should include, in its main body, a generic introduction to IOA. This might be moved to an annex, or the reader might be directed to other publications. This would free up space for a description of the method that was actually used for this analysis. What indicators were used? In what units? What calculations were performed? Were all data points consistent across the entire time series? How was inflation addressed? What about exchange rates from local currencies to the currency of the model (€?)?	→ Thank you. We have kept the generic explanation and have added sub-sections that describe in more detail the methods used in this analysis, including related to time units, monetary units, currency conversions, inflation, etc.
An annex detailing the countries (or regions) considered to represent the Global North and South for the purposes of this research would be helpful, unless "the wealthy economies of Europe" means the 28 EU member states (then please state that explicitly)? Is my understanding correct that the United Kingdom, Switzerland, and Norway are not included in this analysis? Maybe a clarifying list would be really helpful.	→ Thank you. We have included a full list in the SI.
What are the implications for the analysis of data on (some) of the countries in the Global North being available in national detail and the countries of the Global South only as part of regional groupings?	→ Thank you for this comment. We have now discussed this in the methods section.
80: What are supply chain networks and how are they captured in EEMRIO models?	→ Supply chain networks refers to the process by which products are produced with inputs from various production plants dispersed around the world. EEMRIO tables look at supply and use from a product or industry point of view, displaying the internal relationships between industries in an economy in relation to the production and consumption of their products and the products that are imported from elsewhere.
81: In what sense does the 'supply chain network' reach the final consumer?	→ What we capture here is the labour embodied in the products that reach the final consumer, regardless of where in the world the various inputs to the product were produced and in what order.
Table 1: Could you provide more detail on what 'elementary occupations' and professionals are? I am not familiar with the terms and they are not self-explanatory.	→ According to the International Standard Classification of Occupations (ISCO) by ILO, 'elementary occupations' involve the performance of simple tasks which may require the use of hand-held tools and physical effort. Tasks can be, for example, cleaning, restocking supplies and performing basic maintenance in buildings; washing cars and windows; performing various simple farming, fishing etc. Elementary occupations are classified into the following subgroups: - Cleaners and Helpers - Agricultural, Forestry and Fishery Labourers

	 - Labourers in Mining, Construction, Manufacturing and Transport - Food Preparation Assistants - Street and Related Sales and Services Workers - Refuse Workers and Other Elementary Workers We have added a brief description to the text in the Methods section along with a reference to further documentation that describes all occupations.
Due to the lack of detail in the description of the method and the data (in the labor extension of the MRIO model, in particular), it is quite difficult for me to interpret the results. In order to interpret the results, it would be additionally helpful to have data on the population of the Global North and South (as defined for this study) and their age distribution.	→ We have revised the Table in the results to show total workers employed and total labour time rendered, together with hours per worker. This data is internal to EXIOBASE and allows us to avoid relying on external sources that may not align with the specific set of countries represented in EXIOBASE.
This is also actually a methods question, but the results consistently refer to traded goods. Does this include services?	→ Yes, it refers to both goods and services. We have revised to specify this at first mention.
Again, probably methods-related: There is a jump in Global South low-skilled labor between 2006 and 2007 that seems to be 'compensated' by a drop in medium-skilled labor, making this look like a change in classification. Can you confirm or otherwise comment?	→ This jump is present in EXIOBASE and we do not know why it occurs, it is not explained.
Table 2: What's the reasoning for normalizing labor hours by the total population rather than by working-age population? Why do the changes observed mean that more population is entering the workforce? Couldn't they also mean that the same (or fewer people) are working more? What indicator provides the number of workers per country?	→ Thank you for these questions. We have revised the table to include data only on number of workers employed, number of hours worked, and hours per worker.
Figure 2: It might be informative to show these results for the Global North as well? So, to show net-flows of embodied labor for each of the two regions? Because, as far as I understand, the export flows in Figure 2 could also be flows from the Global South to the Global South, right? I see that this data is presented in Figures 3 and 4. It might be useful to make the relation, especially between Figures 2 and 4 clear: the darker gray segments of the bars in 4 correspond to the net-exports in 2?	→ The results for the global North will be exactly the opposite as Fig 2. In other words, the South's imports are the North's exports, and vice versa. We are only tracing flows here between the North and South. Fig 3 simply shows the net South-North flow. It is a net-appropriation for the North, and a net-drain from the South (the same quantity in each case). We have edited the captions to indicate the correspondence between the images.
Figure 3 (methods-related): how was net-appropriation calculated? As labor embodied in exports minus labor embodied in imports? How were re-exports accounted for?	→ Yes, net-appropriation was calculated as labour embodied in exports minus labour embodied in imports. These are final footprint metrics, so they account for re-exports. We have added this to the introduction and methods where appropriate.
Methods-related: Did you account for the relative purchasing-power of wages at all? This partially	→ We have revised the methods section to include a discussion of the currency units. EXIOBASE uses

relates to currency conversion, which is not discussed in the methods chapter!	MER, which is superior to PPP when it comes to measuring purchasing power in the international market.
I understand that your argument goes more strongly into the direction of ‘these are the wages that would have had to be paid in the North if workers there had done the work themselves’; it might be useful to make this argument (and what it does and does not capture) more explicit.	→ We have now discussed this further in the SI.
253: Could you provide an explanation as to why the focus on traded goods here ‘maximizes compatibility’? Isn’t part of the argument around the world-systems that the periphery actually produces distinctly different goods for export than do the countries of the core?	→ Here we are focusing on wages paid to workers involved in production of traded goods, which is relevant to unequal exchange (i.e., we are not assessing wages paid to workers involved in domestic or subsistence production, which in the global South tend to be lower). We have edited to remove the phrase “to maximize comparability” in order to avoid confusion.
251-260: In terms of the growth rate (not in absolute terms, as you point out), wages in the South grew much more dramatically (+250%) than in the North (98%), which has to do with the low level, of course, but might be worth watching out for in the wording.	→ We have edited to clarify that Southern wage gains have not matched Northern wage gains in absolute terms. Yes, while the absolute gap has increased, relative inequality has decreased. We address the latter in Figure 7 and accompanying text.
Figure 6: I’m still not clear about why purchasing power is not relevant here, both between the North and the South and within each of the regions over time. Do 2005 Euros have the same purchasing power in 2020 as they did in 2005? Doesn’t this matter for our interpretation of wage increases?	→ Yes, constant 2005 Euros is a “real” metric, in other words, corrected for price changes over time. Note that this metric is used to assess relative command over products in the global market (i.e., internationally traded goods), not over domestic goods. For the latter, yes, one 2005 euro will have more purchasing power in some countries than in others. This is addressed by PPP, which is not relevant to the international inequalities we are assessing here. We have discussed this in the methods.
Methods-related: Could you please also describe, in the methods chapter, the particular approach to GDP calculation used here? I’m more used to seeing the relative power of labor discussed in terms of income from labor vs income from capital/wealth and it would be great to know what the remaining 50% or so of GDP consist of in this case. It would be the value added approach? So that in the Global South, labor is somewhat more strongly exploited in a Marxian sense than in the North?	→ Labour share of GDP is a very common metric, used by many international agencies. In private production, the income that is not allocated to labour is allocated to profits/capital. However not all labour here is in private production. Yes, the implication is that labour is somewhat more exploited in the global South, within the national economy.
The argument in the paper would be more compelling if the discussion focused on the results presented rather than bringing in a new topic (wage inequality) that is not really covered in the results. Similar concern applies to the discussion of the role of technology and the MER vs. PPP aspect, which – you know from my questions – I have been wondering about, but which is not raised in the methods section.	→ Thank you for this point. We have re-written the Discussion section so that it focuses more directly on the results of the paper.

REVIEWER 3	RESPONSES
1) The author has the right to focus on unequal exchange but should make clear that other dynamics obtained in the historical constitution of cores and peripheries including inter alia settler colonial land alienation/genocide during the mercantilist and early imperial period; and violent political engineering (coup d'état etc) and sanctions during the later colonial and neo-colonial periods; Bagchi and Kadri are essential here	→ Thank you for this comment. We have revised the introduction to explain more about the constitution of the world-system, with citations of both Bagchi and Kadri.
2) Theorists of unequal exchange from Marini to Amin to Alrghiri Emmanuel to all work on the North African region make clear that the issue is not "raw" commodities versus industry; this is clear in the preface fo Emmanuel but also later work on export-oriented industrialization; and the even later work of e.g. Intan Suwandi and John Smith; the authors' discussion of this is muddled	→ We have now assessed wage inequalities and unequal exchange within sectors, finding that large North-South wage gaps and net South-North flows prevail across all sectors, and have revised the text in several places to indicate that our results confirm arguments made by Marini/Amin/etc that sectoral differences are not the issue. We have also now mentioned that the growing wage gaps have occurred despite the increase in industrial manufacturing as a share of the South's total production and exports (as per Suwandi and Smith).
3) There is no gender analysis; of course uncompensated household labor (or broader social reproduction) is not visible in these statistics but that must be noted; see for example the work of Naidu and Ossome on the agrarian question of gendered labor;	→ We have revised to note this point.
4) On p 11 line 324 this would accordingly call for revising the judgment: agricultural labor is not adequately accounted for in the above statistics	→ We have removed this section from the discussion.
5) The line about internal peripheries seems a bit throw-away; the dependency analysis is premised on labor reservoirs which themselves are implicitly 'internal' peripheries	→ We have edited to reflect this.

Reviewers' Comments:

Reviewer #1:

Remarks to the Author:

The revised manuscript substantially addresses the reviewers' comments. I am pleased to recommend acceptance of the manuscript as it extends the empirical literature on unequal exchange, particularly through the disaggregated analysis of different skill levels of workers. Just a few comments on the author's response.

1) In the new version, it is more clearly stated that the estimates of the unequal exchange in hours worked are made under the assumption of homogeneous labour with identical productivity for all countries. Given this assumption, it is tacitly acknowledged that the estimates presented in the paper represent the maximum possible size of the unequal exchange of hours worked between the global North and South. In reality, it is likely to be smaller. The authors specify that this is a second-best choice resulting from the lack of appropriate data and methods to calculate international differences in physical productivity. I would just like to point out that there are, however, proposals in the literature based on the use of real value added in PPP, based on average world prices rather than US prices, as a possible method to homogenise labour input internationally (Reich 2007, 2014; Ricci 2019, 2021). In other words, value added in PPPs could be considered as a proxy for net physical product. While it is true, as the authors rightly point out, that PPPs are not appropriate for analysing international differences in monetary income, they can be used to measure international differences in real value.

2) The revised version of the paper specifies that the estimation of monetary transfers is done by considering only the value added appropriate to wages and not total value added. As such, it is a partial estimate that does not take into account the absolute differences between global North and South capital income per unit of labour.

Andrea Ricci

Reviewer #3:

Remarks to the Author:

authors have satisfactorily replied to critique